# Bruxism Throughout the Lifespan and Variants in *MMP2*, *MMP9* and *COMT*

**DOI:** 10.3390/jpm10020044

**Published:** 2020-05-27

**Authors:** Alexandre R. Vieira, Rafaela Scariot, Jennifer T. Gerber, Juliana Arid, Erika C. Küchler, Aline M. Sebastiani, Marcelo Palinkas, Kranya V. Díaz-Serrano, Carolina P. Torres, Simone C. H. Regalo, Paulo Nelson-Filho, Diego G. Bussaneli, Kathleen Deeley, Adriana Modesto

**Affiliations:** 1Department of Oral Biology, University of Pittsburgh School of Dental Medicine, 412 Salk Pavilion, 335 Sutherland Drive, Pittsburgh, PA 15261, USA; bussaneli@gmail.com (D.G.B.); kbd1@pitt.edu (K.D.); ams208@pitt.edu (A.M.); 2Department of Pediatric Dentistry, University of Pittsburgh School of Dental Medicine, Pittsburgh, PA 15213, USA; 3Department of Oral and Maxillofacial Surgery, Positivo University, Curitiba, PR 81280-330, Brazil; rafaela_scariot@yahoo.com.br (R.S.); jennifergerber96@gmail.com (J.T.G.); sebastiani.aline@gmail.com (A.M.S.); 4Department of Pediatric Dentistry, USP, Ribeirão Preto, SP 14040-904, Brazil; juliana.arid@usp.br (J.A.); erikacalvano@gmail.com (E.C.K.); palinkas@usp.br (M.P.); dkranya@forp.usp.br (K.V.D.-S.); secretaria807@forp.usp.br (C.P.T.); nelson@forp.usp.br (P.N.-F.); 5Department of Morphology, Physiology and Basic Pathology, USP, Ribeirão Preto, SP 14040-904, Brazil; simone@forp.usp.br; 6Department of Pediatric Dentistry, UNESP, Araraquara, SP 14801-385, Brazil

**Keywords:** biomarkers, temporomandibular disorders, genetics, matrix metalloproteinases, child dentistry, oral diagnosis

## Abstract

Bruxism is a masticatory muscle activity characterized by grinding of the teeth and clenching of the jaw that causes tooth wear and breakage, temporomandibular joint disorders, muscle pain, and headache. Bruxism occurs in both adults and children. Clinical characteristics and habits were evaluated in an adult sample. Moreover, we used DNA samples from 349 adults and 151 children to determine the presence of association with specific genes. Genomic DNA was obtained from saliva. The markers *rs2241145* and *rs243832* (metalloproteinase 2 (*MMP2*)), *rs13925* and *rs2236416* (metalloproteinase 9 (*MMP9*)), and *rs6269* (cathecol-o-methyltransferase (*COMT*)) were genotyped. Data were submitted to statistical analysis with a significance level of 0.05. In adults, in univariate logistic regression, presence of caries, attrition, and use of alcohol were increased in bruxism individuals (*p* < 0.05). In addition, in adults, there was an association between bruxism and *MMP9* (*rs13925*, *p* = 0.0001) and bruxism and *COMT* (*rs6269*, *p* = 0.003). In children, a borderline association was observed for *MMP9* (*rs2236416*, *p* = 0.08). When we performed multivariate logistic regression analyses in adults, the same clinical characteristics remained associated with bruxism, and orthodontic treatment was also associated, besides *rs13925*, in the AG genotype (*p* = 0.015, OR_a_: 3.40 (1.27–9.07)). For the first time, we provide statistical evidence that these genes are associate with bruxism.

## 1. Introduction

Bruxism is considered a repetitive jaw-muscle activity characterized by clenching or grinding of the teeth and/or by bracing or trusting of the mandible [1]. This activity can occur while the individual is awake or asleep, and tends to decline with age [2], affecting as many as 20% of adults [1]. The reported prevalence in children is highly variable from 3.5 to 40.6% [3]. Bruxism can be attributed to peripheral (morphological) factors but it is mainly centrally regulated [4]. Psychological factors seem play an important role in bruxism [5,6]. It is seen in individuals with stress and anxiety, as well as individuals with underlying neurodevelopmental disorders such as Huntington’s disease [7,8], Rett syndrome [9], Angelman syndrome [10], autism spectrum disorders [11], and Down syndrome [12]. Evidence from twin studies suggests a genetic additive component that accounts for as much as 52% of the variation seen among affected individuals [13,14], and include nicotine dependence as an etiological factor [15]. Motivated by the evidence of a genetic component to bruxism, association studies with common genetic variants have been conducted, suggesting an association exists between genes and bruxism (dopamine receptor D genes *DRD2*, *DRD3*, *DRD5,* and the 5-hydroxytryptamine receptor 2A gene, *HTR2A)* [16,17,18].

Metalloproteinase 2 (*MMP2*) and the metalloproteinase 9 (*MMP9*) are gelatinases associated with the lysis of type IV collagen and elastin. *MMP9* is implicated in a number of neurodegenerative disorders [19] and is associated with stress conditions [20], which is possibly associated with bruxism [21]. Besides studying the serotonin [17] and dopamine [16], pathways to identify genetic variants contributing to bruxism is a sensible choice due to the evidence that neurotransmitters in the central nervous system are involved with bruxism. Cathecol-o-methyltransferase (*COMT*) is one of many enzymes that degrade catecholamines [22]. 

Here, we tested for two metalloproteinases and a catecholamine degradant associated with dental clinical characteristics to identify factors that may be useful to determine the etiology or potential consequences of bruxism, which may provide a justification for alternative clinical management. Our hypothesis was that *MMP2*, *MMP9*, and/or *COMT* associate with bruxism, and that variation in these genes may be biomarkers of the risk of developing the condition.

## 2. Subjects and Methods

### 2.1. Ethical Aspects

The adult participants were selected from the database of Dental Registry and DNA Repository (DRDR), University of Pittsburgh School of Dental Medicine. Starting in September of 2006, all individuals that sought treatment at the university were invited to be part of the registry. All these individuals gave written informed consent, authorizing the extraction of information from their dental records, as well as providing a saliva sample (University of Pittsburgh Institutional Review Board (IRB) approval #0606091) [23]. When possible, 4 mL of unstimulated whole saliva was collected from each participant. The child participants were selected from the Pediatric Dentistry Clinic and the Service of Bruxism and Temporomandibular Disorder in Childhood (SABDI), Department of Pediatric Dentistry of the School of Dentistry of Ribeirão Preto, University of São Paulo. Informed consent was obtained from all participating children or parents/legal guardians (University of São Paulo Ethics Committee approval #35323314.7.0000.5419 and University of Pittsburgh Institutional Review Board (IRB) approval #0511110). 

### 2.2. Sample Selection

**Adults:** We included 349 individuals that were selected from 5486 participants of the DRDR database. First, we identified all individuals that had a diagnostic code of self-reported bruxism. A total of 69 individuals (1.24% of the total individuals in the study, 34 females and 35 males, mean age of 41.33 years, 52 Whites, 10 Blacks, 7 others) were identified, which suggests that bruxism is under-reported in our Dental Registry and DNA Repository project, as we expected the frequency of bruxism to be at least 8% [2]. All subjects were described as grinding their teeth during their sleep. Then, for each individual with bruxism, we identified four individuals with no bruxism to serve as a comparison, matching them by age, sex, and ethnicity, in order to increase the power of the study. A total of 280 individuals were selected to compose the comparison group (135 females, 145 males, mean age 41.285 years, 212 Whites, 40 Blacks, 27 others). A number of dental characteristics were obtained from the subjects’ records, such as caries, gingival resection, dental fracture, attrition, temporomandibular joint dysfunction (TMD), dental restorations, history of prosthodontic treatment, history of orthodontic treatment, and partial edentulism. In addition, we evaluated the habits from the individuals of the sample in terms of use of alcohol, tobacco, and/or drugs. Patients who smoked at least one cigarette daily were considered smokers. For alcohol, the criterion utilized to determine its recurring use were people who drank at least every week. Additionally, the individuals that were using illicit drugs were considered users of drugs. All these data were categorized into dichotomized continuous variables for statistical analysis. 

**Children**: We involved 151 children (81 males and 70 females, mean age 9.34 years) in this study. Of this total, 61 had sleep bruxism (33 males and 28 females). Patients were diagnosed with probable bruxism, according to the International consensus on the assessment of bruxism, 2018, based on the parents’ report referring to audible tooth grinding sounds or tooth clenching during sleep, frequent headaches and orofacial pain on awakening, chewing and/or opening the mouth. In addition, signs and symptoms related to bruxism, such as dental wear facets and fractures of restorations; dental impressions on the cheek mucosa and tongue, tenderness and pain on the temporal and masseter muscles during bilateral palpation, were evaluated.

### 2.3. Genetical Analyses

The markers *rs2241145* and *rs243832* (*MMP2*), *rs13925* and *rs2236416* (*MMP9*), and *rs6269* (*COMT*) were genotyped. The *MMP2* gene location is in the long arm (q) of chromosome 16, at position 12.2; the *MMP9* gene location is in the long arm (q) of chromosome 20, at position 13.12; and the *COMT* gene location is in the long arm (q) of chromosome 22, at position 11.21 (Table 1). Genomic DNA samples were obtained from saliva. Genotyping was performed by the Taqman method [24], with a QuantStudio 6 Flex instrument and pre-designed probes (Applied Biosystems, Foster City, CA, USA). 

### 2.4. Statistical Analyses

The dependent variable was the presence or absence of bruxism, which was compared with independent variables. The clinical characteristics were considered as presence or absence. To analyze the association with the clinical characteristic with the dependent variable, we used univariate logistic regression. The genotyping data were analyzed by chi-squared or Fisher’s exact test for both markers. For the multiple logistic regression model, it the independent variables whose relation with bruxism presented as *p* < 0.2 were added. In the final model, we only used the independent variables whose *p*-value remained ≤ 0.05, or those that needed to adjust the other variables. Data were analyzed through the software SPSS (Statistical Package for Social Science, V.20, IBM, Armonk, NY, USA) and in PLINK [25].

## 3. Results

Adults individuals with bruxism showed more frequency of caries (*p* = 0.001, OR: 2.50 (1.46–4.30)) and dental attrition (*p* < 0.001, OR: 3.40 (1.82–6.34)). Regarding the frequency of habits of the individuals in the sample, there was a difference in alcohol consumption between groups. The bruxism group reported consuming more alcohol (*p* = 0.011, OR: 2.39 (1.22–4.71)); however, for tobacco and drug consumption, there was no statistical difference (*p* > 0.05). These data are shown in Table 2 in full.

All genotypes were in Hardy–Weinberg equilibrium. In adults, there was an association between bruxism and *MMP9* (*rs13925*, *p* = 0.0001) and bruxism and *COMT* (*rs6269*, *p* = 0.003) in genotype evaluation. In children, a borderline association was observed for *MMP9* (*rs2236416*, *p* = 0.08) (Table 3). 

## 4. Discussion

As expected, adults diagnosed with bruxism had higher caries experience (likely related to excessive restorative needs, which can inflate caries experience scores) and dental attrition. In the multivariate analysis, having had orthodontic treatment increased the chance to develop bruxism [26]. This can be explained by the fact that people seeking orthodontic treatment have more malocclusion, which is a suggested risk factor for bruxism [26]. One association that did not appear in our study was the association between bruxism and TMD. One of the possible causes for this may be the fact that both bruxism and TMD were reported by the patient and not by diagnostic criteria such as the RDC-TMD (Research Diagnostic Criteria for Temporomandibular Disorders) [27], which is highly sensitive. We also found that individuals with bruxism were more likely to use alcohol. There are some studies associating the use of medications and drugs to bruxism [28,29]. We believe these data corroborate the suggestion that individual susceptibility to become addicted may be underlying the etiology of bruxism.

*MMP9* has a role in the development of sensory circuits during early postnatal life, and misregulated activation of this enzyme is implicated in a number of neurodegenerative disorders, including traumatic brain injury, multiple sclerosis, and Alzheimer’s disease [19]. In mice, *Mmp9* is differently expressed in the hypothalamus under stress conditions [20]. *Mmp9* expression is also increased at 12 h after mild treadmill exercise in rat hippocampus [30]. *Mmp2* activity did not change in rat hippocampus after treadmill [30]; however, Mmp2 protein is detected in grafts of olfactory ensheathing cells that elicit robust axonal regeneration in the injured spinal cord of adult rats in vivo [31]. In aggregate, the evidence from mice that *MMP9* expression changes under stress and our results showing that the gene associates with bruxism provide support to the idea that *MMP9* genetic variants may predispose individuals, once exposed to stressful conditions, to clench or grind their teeth. The gene is essential for an adequate inflammatory response and tissue repair [32]. Genetic variants in *MMP9* can alter its function by increasing promoter activity or altering its binding abilities [33]. We can argue for a potential direct effect of *MMP9* genetic variants in the reaction of adults and children exposed to stressful situations, manifested by grinding their teeth. 

The association we found between genetic variants in *COMT* and bruxism, only in the univariate analysis, could be explained because the marker *COMT* has been associated with psychological and emotional changes, and moreover it has influence on the interactions between the prefrontal cortex and the limbic system, which is responsible for dopamine and serotonin modulation [34]. 

Other aspects that may explain the lack of association in children are the multifactorial nature of bruxism etiology and possible specific factors, which make bruxism more common in children than in adults. Systematic reviews have reported highly variable values of prevalence of bruxism in children, ranging from 3.5% to 40.6% [3] and 15% to 50% [35]. These data make us suggest that other genetic aspects are involved in the etiology of bruxism in children. It is a limitation of our study that we could not evaluate the presence of stress markers, of any possible confounding effects from the use of psychoactive drugs, or of underlying psychiatric conditions. Future analyses can also test the role of vitamins and other mediators, as well as circulating levels of progenitor cells, which appear to modulate the risk for a number of oral conditions such as periodontitis and tooth loss [36,37].

In summary, we have provided statistical evidence for the role of *MMP9* in self-reported bruxism in adults and suggest that bruxism is the result of stress and could be detected early if surrogates of stress are evaluated.

## Figures and Tables

**Table 1 jpm-10-00044-t001:** Genes and polymorphisms studied.

Gene	Position	Polymorphism	MAF	Base Change
*MMP2*	16q12.2	rs2241145	0.499	C/G
		rs243832	0.467	C/G
*MMP9*	20q13.12	rs13925	0.148	A/G
		rs2236416	0.159	A/G
*COMT*	22q11.21	rs6269	0.356	A/G

Notes: MAF means minor allele frequency. Obtained from database: ncbi.nlm.nih.gov. MMP means matrix metalloproteinase. COMT means catechol-O-methyltransferase.

**Table 2 jpm-10-00044-t002:** Comparison of dental characteristics and habits between adults with bruxism (cases) and individuals without bruxism (control).

Variables		Case N (%)	Control N (%)	*p*-Value	OR (95% CI)
Caries	Present	37 (54.4)	89 (32.2)	**0.001**	2.50 (1.46–4.30)
	Absent	31 (45.6)	187 (67.8)	reference	-
Gingival resection	Present	6 (8.8)	20 (7.2)	0.660	1.23 (0.47–3.21)
	Absent	62 (91.2)	256 (92.8)	reference	-
Dental fracture	Present	7 (10.3)	25 (9.1)	0.753	1.15 (0.47–2.78)
	Absent	61 (89.7)	251 (90.9)	reference	-
Attrition	Present	22 (32.4)	34 (12.3)	**<0.001**	3.40 (1.82–6.34)
	Absent	46 (67.6)	242 (87.7)	reference	-
Temporomandibular joint dysfunction (TMD)	Present	33 (48.5)	102 (37)	<0.082	1.60 (0.94–2.64)
	Absent	35 (51.5)	174 (63)	reference	-
Dental restorations	Present	61 (89.7)	213 (79.5)	0.057	2.25 (0.97–5.19)
	Absent	7 (10.3)	55 (20.5)	reference	-
Prosthodontics	Present	31 (46.3)	118 (44.2)	0.760	1.80 (0.63–1.86)
	Absent	36 (53.7)	149 (55.8)	reference	-
Orthodontia	Present	10 (14.9)	25 (9.3)	0.184	1.70 (0.77–3.75)
	Absent	57 (85.1)	243 (90.7)	reference	-
Partial edentulism	Present	16 (23.5)	71 (25.7)	0.709	0.88 (0.47–1.65)
	Absent	52 (76.5)	205 (74.3)	reference	-
Alcohol use	Present	16 (23.9)	31 (11.6)	**0.011**	2.39 (1.22–4.71)
	Absent	51 (76.1)	237 (88.4)	reference	-
Tobacco use	Present	13 (19.1)	80 (29.7)	0.083	0.55 (0.28–1.07)
	Absent	55 (80.9)	189 (70.3)	reference	-
Drug use	Present	9 (13.2)	38 (14.2)	0.833	0.91 (0.42–2.00)
	Absent	59 (86.8)	229 (85.8)	reference	-

Note: Univariate logistic regression, with confidence level of 0.05. OR: odds ratio and CI: confidence interval. Bold indicates statistically significant *p*-values.

**Table 3 jpm-10-00044-t003:** Summary of the association results.

Gene/ Marker		Alleles	Case n (%)	Control n (%)	*p*-Value	Genotypes	Case n (%)	Control n (%)	*p*-Value
*MMP2* *rs2241145*	**Adults**	C	39 (20.1)	155 (79.9)	0.91	CC	3 (11.1)	24 (88.9)	0.28
		G	75 (20.5)	291 (79.5)		CG	33 (23.5)	107 (76.5)	
						GG	21 (18.6)	92 (81.4)	
	**Children**	C	58 (38.9)	91 (61.1)	0.71	CC	13 (36.1)	23 (63.9)	0.86
		G	62 (41.1)	89 (58.9)		CG	32 (41.5)	45 (58.5)	
						GG	15 (40.5)	22 (59.5)	
*MMP2 rs243832*	**Adults**	C	72 (23)	241 (77)	0.22	CC	21 (26.9)	57 (73.1)	0.34
		G	64 (19.1)	271 (80.9)		CG	30 (19.1)	127 (80.9)	
						GG	17 (19.1)	72 (80.9)	
	**Children**	C	58 (37.2)	98 (62.8)	0.30	CC	13 (35.1)	24 (64.9)	0.52
		G	62 (43)	82 (57)		CG	32 (39)	50 (61)	
						GG	15 (48.4)	16 (51.6)	
*MMP9 rs2236416*	**Adults**	A	16 (19.1)	68 (80.9)	0.83	AA	1 (20)	4 (80)	0.96
		G	102 (20.1)	406 (79.9)		AG	14 (18.9)	60 (81.1)	
						GG	44 (20.3)	173 (79.7)	
	**Children**	A	101 (40.1)	151 (59.9)	0.48	AA	41 (37.6)	68 (62.4)	0.08
		G	21 (45.6)	25 (54.4)		AG	19 (55.9)	15 (44.1)	
						GG	1 (16.7)	5 (83.3)	
*MMP9 rs13925*	**Adults**	A	19 (14.4)	113 (85.6)	**0.04**	AA	3 (5.70)	49 (94.3)	**0.0001**
		G	111 (22.4)	385 (77.6)		AG	13 (46.4)	15 (53.6)	
						GG	49 (20.9)	185 (79.1)	
	**Children**	A	21 (51.2)	20 (48.8)	0.12	AA	1 (33.3)	2 (66.6)	0.15
		G	99 (38.5)	158 (61.5)		AG	19 (54.3)	16 (45,7)	
						GG	40 (36.1)	71 (63.9)	
*COMT rs6269*	**Adults**	A	44 (18.4)	195 (81.6)	0.36	AA	14 (27.5)	37 (72.5)	**0.003**
		G	80 (21.4)	293 (78.6)		AG	16 (11.7)	121 88.3)	
						GG	32 (27.1)	86 (72.9)	
	**Children**	A	52 (43.3)	68 (56.7)	0.63	AA	13 (56.5)	10 (43.5)	0.21
		G	76 (41.3)	108 (58.4)		AG	26 (36.1)	46 (63.9)	
						GG	25 (44.6)	31 (55.4)	

In the multiple regression analyses of the adult sample, there was an association with presence of caries [*p* < 0.001, OR_a_: 3.68 (1.83–7.40)], dental attrition [*p* < 0.001, OR_a_: 4.28 (1.97–9.28)], and orthodontic treatment [*p* < 0.001, OR_a_ 4.70 (2.35–9.40)]. Frequency of habits was increased among adults who used alcohol [*p* < 0.025, OR_a_ 2.95 (1.14–7.65)]. The only gene that was associated in the multiple regression analyses was the *MMP9* (*rs13925*) with AG genotype [*p* = 0.015, OR_a_: 3.40 (1.27–9.07)] (Table 4). Bold indicates statistically significant *p*-values.

**Table 4 jpm-10-00044-t004:** Multivariate regression analyses in adults, including only variables with *p* < 0.2, in univariate analyses.

Variables		*p_a_* Value	OR_a_
Caries	Present	**<0.001**	3.68 (1.83–7.40)
	Absent	reference	-
Attrition	Present	**<0.001**	4.28 (1.97–9.28)
	Absent	reference	
Temporomandibular joint dysfunction (TMD)	Present	0.260	1.48 (0.74–2.93)
	Absent	reference	-
Dental restoration	Present	0.875	1.07 (0.44–2.57)
	Absent	reference	-
Orthodontics	Present	**0.014**	3.24 (1.26–8.29)
	Absent	reference	
Alcohol use	Present	**0.025**	2.95 (1.14–7.65)
	Absent	reference	
Tobacco use	Present	0.087	0.47 (0.20–1.11)
	Absent	reference	
*MMP9* *rs13925*	GG	reference	
	AG	**0.015**	3.40 (1.27–9.07)
	AA	0.081	0.27 (0.06–1.17)
*COMT* *rs6269*	CC	reference	
	AG	0.051	0.38 (0.14–1.00)
	AA	0.98	0.42 (0.29–2.37)

Note: Multivariate logistic regression with confidence level of 0.05. ORa: odds ratio adjusted. Bold indicates statistically significant *p*-values.

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
