# Peer review of "Bruxism Throughout the Lifespan and Variants in MMP2, MMP9 and COMT"

_jpm, 2020, doi:10.3390/jpm10020044_

Round 1

Reviewer 1 Report

The Authors tackle an interesting topic, but the manuscript lacks some information. The list is below:

  1. TMD - there is no abbreviation
  2. There is absolutely no information on how the sample was taken. How was saliva collected, what type of saliva, volume and did the probate have to prepare for collection in any way? Only the publication is given, but there should be a brief information on sampling
  3. Line 98 - is it a drug in the sense of psychoactive substances? Have psychiatric treatment and the use of psychoactive drugs been considered? Or the drugs were used as stimulants?

Author Response

The reviewer had the following concerns:

  • TMD - there is no abbreviation

RESPONSE: We added the definition the first time the abbreviation appears as suggested.

  • There is absolutely no information on how the sample was taken. How was saliva collected, what type of saliva, volume and did the probate have to prepare for collection in any way? Only the publication is given, but there should be a brief information on sampling

RESPONSE: We added the information as requested.

  • Line 98 - is it a drug in the sense of psychoactive substances? Have psychiatric treatment and the use of psychoactive drugs been considered? Or the drugs were used as stimulants?

RESPONSE: Yes, in the sense of psychoactive substances. In these analyses, we did not separately considered psychoactive drugs.

Reviewer 2 Report

Authors evaluated the association of various gene markers with bruxism in both adults and children. The manuscript is short and succinct.

COMT spelling error in Page 2 Line 61.

Page 4 line 103: Suggest replacing the word "USED" with Included.

Page 4: Line 123 in Statistical analysis: Kindly rephrase the sentence starting "It remained in the final model........" to provide more clarity. 

Page 5: Suggest rephrasing the line "There was a statistical difference regarding the habits...." to provide better clarity for the readers. 

Discussion: 

Authors have eluded that the difference in the association of markers between adults and children (non-significant) to the etiology factors. Does this mean these markers are related to the etiology rather than bruxism?. Do they have additional data on stress related psychological issues like anxiety or depression?

Author Response

The reviewer had the following concerns:

COMT spelling error in Page 2 Line 61.

RESPONSE: We made the correction requested.

Page 4 line 103: Suggest replacing the word "USED" with Included.

RESPONSE: We made the suggested change.

Page 4: Line 123 in Statistical analysis: Kindly rephrase the sentence starting "It remained in the final model........" to provide more clarity. 

RESPONSE: We rewrote the sentence as suggested.

Page 5: Suggest rephrasing the line "There was a statistical difference regarding the habits...." to provide better clarity for the readers.

RESPONSE: We rewrote the text as suggested.

Discussion: 

Authors have eluded that the difference in the association of markers between adults and children (non-significant) to the etiology factors. Does this mean these markers are related to the etiology rather than bruxism?. Do they have additional data on stress related psychological issues like anxiety or depression?

RESPONSE: This is an important point and we added some language to highlight this limitation, lack of stress data, in the discussion section.